# Suspended Sediment Modeling Using a Heuristic Regression Method Hybridized with Kmeans Clustering

**Rana Muhammad Adnan** [1], **Kulwinder Singh Parmar** [2], **Salim Heddam** [3], **Shamsuddin Shahid** [4] and **Ozgur Kisi** [4,5,*]

1 State Key Laboratory of Hydrology-Water Resources and Hydraulic Engineering, Hohai University, Nanjing 210098, China; rana@hhu.edu.cn
2 Department of Mathematics, IKG Punjab Technical University, Jalandhar, Kapurthala 144603, India; kulmaths@gmail.com
3 Agronomy Department, Hydraulics Division, Faculty of Science, University of Skikda, Skikda 21000, Algeria; heddamsalim@yahoo.fr
4 Faculty of Engineering, School of Civil Engineering, Universiti Teknologi Malaysia (UTM), Johor Bahru 81310, Malaysia; sshahid@utm.my
5 School of Technology, Ilia State University, Tbilisi 0162, Georgia
* Correspondence: ozgur.kisi@iliauni.edu.ge

**Abstract:** The accurate estimation of suspended sediments (SSs) carries significance in determining the volume of dam storage, river carrying capacity, pollution susceptibility, soil erosion potential, aquatic ecological impacts, and the design and operation of hydraulic structures. The presented study proposes a new method for accurately estimating daily SSs using antecedent discharge and sediment information. The novel method is developed by hybridizing the multivariate adaptive regression spline (MARS) and the Kmeans clustering algorithm (MARS–KM). The proposed method's efficacy is established by comparing its performance with the adaptive neuro-fuzzy system (ANFIS), MARS, and M5 tree (M5Tree) models in predicting SSs at two stations situated on the Yangtze River of China, according to the three assessment measurements, RMSE, MAE, and NSE. Two modeling scenarios are employed; data are divided into 50–50% for model training and testing in the first scenario, and the training and test data sets are swapped in the second scenario. In Guangyuan Station, the MARS–KM showed a performance improvement compared to ANFIS, MARS, and M5Tree methods in term of RMSE by 39%, 30%, and 18% in the first scenario and by 24%, 22%, and 8% in the second scenario, respectively, while the improvement in RMSE of ANFIS, MARS, and M5Tree was 34%, 26%, and 27% in the first scenario and 7%, 16%, and 6% in the second scenario, respectively, at Beibei Station. Additionally, the MARS–KM models provided much more satisfactory estimates using only discharge values as inputs.

**Keywords:** estimating discharge–sediment relationship; MARS–Kmeans; MARS; ANFIS; M5 model tree

## 1. Introduction

The rapidly growing global population has made freshwater resources scarce and compelled hydrologists to explore methods for better river management and water resource conservation [1]. Accurate modeling of suspended sediment load (SSL) plays a vital role in river restoration, pollution, and soil erosion reduction, thus solving challenges related to water quality, channel design, and the operation of hydraulic structures [2,3]. However, precise forecasting of SSL is challenging due to the concurrent effects of many meteorological and hydrological factors on sediment processes, such as wind speed, evaporation, precipitation, river discharge, water temperature, and ice packs. The variations of these parameters in space and time make the sediment dynamics highly complicated and nonlinear [4,5]. Many SSL estimation models have been developed in the literature, ranging from physically based to data-driven models. Physically based models require a large

volume of different kinds of data and information for reliable estimation of SSL. However, such a large amount of data is difficult to obtain in data-scarce catchments, especially for developing countries [6]. For such cases, data-driven models have demonstrated success in modeling different hydrological phenomena, especially streamflow and sediment load, by capturing the non-stationarity and nonlinear behavior of SSL with fewer data [3,7,8]. Several data-driven modeling approaches, including artificial neural networks (ANN), adaptive neuro-fuzzy inference systems (ANFIS), support vector machines (SVM), the M5 model tree (M5Tree), and multivariate adaptive regression splines (MARS), have shown their efficiency in precise modeling of different hydrological variables [9–12]. Some of those algorithms, such as ANN, have also demonstrated their success in modeling SSL [13].

Different ANN models have been utilized to estimate SSL during the last two decades [13–15]. The studies showed some inherent drawbacks of ANN, including limited regularization and plunging to local minima. The rapid learning and adaptation capacity of ANFIS has made it capable of overcoming the weaknesses of ANN significantly. Therefore, it has been widely employed in recent years for SSL prediction [16–29]. Kisi and Yaseen [16] applied three different ANFIS models, including ANFIS subtractive clustering (ANFIS-SC), ANFIS grid partition (ANFIS-GP), and ANFIS fuzzy c-means (ANFIS-FCM), to predict the suspended sediment concentration of Eel River. Results indicated the satisfactory performance of ANFIS based models in estimating sediment concentration. Bakhtyar et al. [17] and Kabiri-Samani et al. [18] evaluated the prediction accuracy of ANFIS compared to different empirical formulas such as Walton–Bruno (WB), Van Rijn (VR), and CERC in forecasting longshore sediment transport rate (LSTR). They found less error in estimating LSTR using ANFIS models than that obtained using empirical formulas. Mianaei and Keshavarzi [19] examined the ANFIS prediction capability in assessing suspended sediment discharge at the Escanaba River mouth station. They found the ANFIS model's prediction very close to observation. For daily streamflow and sediment discharge estimation at Polavaram and Pathagudem gauging stations of the Godavari catchment, Kumar et al. [20] applied ANFIS and ANN models. As results, they found that the ANFIS model with three membership functions provided the best results for daily streamflow and sediment discharge. Kisi and Kermani [21] utilized the ANFIS-FCM, ANN, and sediment rating curve methods to determine the daily sediment amount at two hydraulic stations operated by the United States Geological Survey. They reported the ANFIS model's capability to improve prediction accuracy by 10% to 16% compared to the ANN model. Vafakhah [22] applied the ANFIS model to sediment load prediction of the Kojor forest watershed near the Caspian Sea using rainfall and streamflow as inputs. He compared the selected method with cokriging (CK), ordinary kriging (OK), and ANN models and reported a better performance of ANFIS compared to CK and OK models. For daily and monthly SSL modeling of the Little Black River and the Great Menderes basin, Rajaee et al. [23] and Firat and Gungor [24] compared the performance of the ANFIS model with ANN and multiple linear regression (MLR). Results demonstrated the dominancy of the ANFIS model over the ANN and MLR models. In dam operation management, the precise calculation of the sediment input to the dam reservoir is very critical. Therefore, Samet et al. [25] used the ANFIS model with ANN and genetic algorithm (GA) models for SSL estimation from temperature, runoff, and CM (three-section method of sediment sampling) data in the Maku dam reservoir of Iran as a case study. Results indicated that the ANFIS model has a "gauss" membership function, which provides more accurate results than the ANN and GA models, with only a 0.968% percentage error. In addition to success in SSL prediction, the ANFIS models also performed well in estimating riverbed load. For forecasting the bed load of three Malaysian rivers (Kurau, Langat, and Muda of Peninsular Malaysia), Chang et al. [26] utilized the ANFIS model. The literature also showed that coupling the wavelet technique with the ANFIS model produces more accurate results than standalone ANFIS models. Mirbagheri et al. [27], Rajaee. [28], and Rajaee et al. [29] applied the ANFIS hybrid model coupled with the wavelet method. They found that the hybrid ANFIS model

provided more accurate results than the standalone ANFIS models in estimating SSL at a USA gauging station.

This study selected the MARS model due to its shorter training process and better ability to model complex nonlinear processes without strong model assumptions than ANN models [30]. Another selected method was the M5Tree model due to its large data handling capability and smaller computational cost than the ANN and SVM models [29]. In recent years, MARS and M5Tree have been applied successfully in modeling runoff and sediment load [31–39]. Malik et al. [37] compared the performance of the MARS model with the SVM-based model (least square SVM) and two ANN models (radial basis and self-organizing map neural network) for estimating daily SSL at different gauging stations in Godavari catchment, India. Results indicated that the radial basis neural network and MARS models provided more satisfactory results than other data-driven models. Senthil Kumar et al. [38] evaluated the accuracy of the M5Tree model in predicting SSL and compared its performance with ANN coupled with backpropagation and Levenberg–Marquardt algorithms, fuzzy logic, and REPTree models. They obtained the most precise sediment concentration simulations using M5Tree.

Although the MARS and M5Tree models demonstrated promising results in previous studies for sediment modeling, both models could not efficiently capture the uncertainties in sediment time series data due to their complex behavior. The literature found that hybrid MARS and M5Tree models can provide more precise results than standalone data-driven models. A hybrid of the wavelet method and the M5Tree model (WM5Tree) was introduced by Goyal et al. [39] to estimate the sediment yield, and they found that the wavelet M5Tree provided more accurate results than ANN models. Nourani et al. [35] endorsed the findings of Goyal et al. [39]. They predicted the daily sediment load of the Lighvanchai and Upper Rio Grande rivers by comparing the WM5Tree with ANN and standalone M5Tree models and found that the hybrid M5Tree model outperformed the other models. Rahgoshay et al. [36] applied M5Tree, MARS, and hybrid of SVM with GA and particle swarm optimization (PSO) models to predict the sediment load of two earth dams. They found that SVM hybrid models (SVM with GA and SVM with PSO) provided more precise results than the MARS and M5Tree models.

The abovementioned literature revealed hybrid models still need improvement for precise modeling of suspended sediment. In this study, a new model is developed through hybridization of MARS with the Kmeans method (MARS–KM) to overcome standalone MARS models' weakness in precisely capturing uncertainties in sediment dynamics. The MARS model and many other machine learning models' main disadvantage is that they are very time-consuming, especially for large amounts of data with high variance, as in sediment load. For this purpose, the K means clustering method is utilized in this study. Therefore, the main contributions of this study are to (1) develop a novel model that introducing the K-means clustering into the MARS model for more accurate and faster estimation; (2) compare the prediction accuracy of the proposed MARS–KMeans model with the commonly used machine learning models, e.g., MARS, M5Tree, and ANFIS in sediment load estimation; and (3) select methods based on 50–50% data division.

K-means has been successfully used in recent literature to improve machine learning models' prediction accuracy due to its robust nature in estimation [40–44]. The application of hybrid the MARS–KMeans method is rarely found in the literature for prediction [45]. There is no published study in the literature that uses the MARS–KM method for modeling any variables in hydrology to the best of our knowledge. This gave impetus to this study.

## 2. Materials and Methods

### 2.1. Case Study and Data Analysis

The Jialing catchment was chosen in this study as a case study due to its vital role in the hydropower operation management of the world's biggest dam, i.e., the Three Gorges Reservoir. The selected catchment is the second largest tributary of the Yangtze River with a drainage area of 160,000 km$^2$ and one of the reservoir's main discharges and

sediment sources. Thus, a precise estimation of sediment load input from the basin to the reservoir is very important for efficient reservoir operation. Two key gauging stations in the Jialing catchment, located at Guangyuan and Beibei, were chosen to predict SSL. These hydraulic gauging stations' geographical locations aid in understanding the overall sediment phenomena and contribution of the catchment to the main Yangtze River. The selected catchment also has two main tributaries, the Qu River on the main channel's left side and the Fu River on the right side (see Figure 1). To estimate the daily sediment load of both gauging stations, the daily data of runoff and suspended sediment from 1 January 2007, to 31 December 2015, were gathered from the Hydrological Yearbooks of the People's Republic of China. The time variation graphs of both stations' sediment loads are illustrated in Figure S1 (see Supplementary Materials). The related authorities checked the reliability and homogeneity of the data before releasing the data. Chinese national standard criteria were followed in obtaining streamflow and sediment measurements. The first vertical profiles (usually 10–30 profiles depending on the river width) are determined for measurement. Then, water depth and velocity of flow (utilizing a velocity meter) at each profile are measured. Flow velocity is recorded for different depths. Water samples for sediment concentrations (SCs) are collected from each depth, and these samples are dried and weighed in the lab. Finally, daily sediment loads are computed by multiplying the SCs by the streamflow [46,47].

Table S1 reports the brief statistics of the used streamflow (Q) and sediment (S) data. In both stations, the Q and S have high skewness, indicating that the time series has many extreme values. An augmented Dickey–Fuller test was applied to the sediment data using the adftest MATLAB command to see if they are stationary or not. For the Guangyuan and Beibei stations, the test statistics were $-27.53$ and $-21.67$, while the critical value was $-1.94$. Test statistics indicate that the sediment data are stationary in both stations.

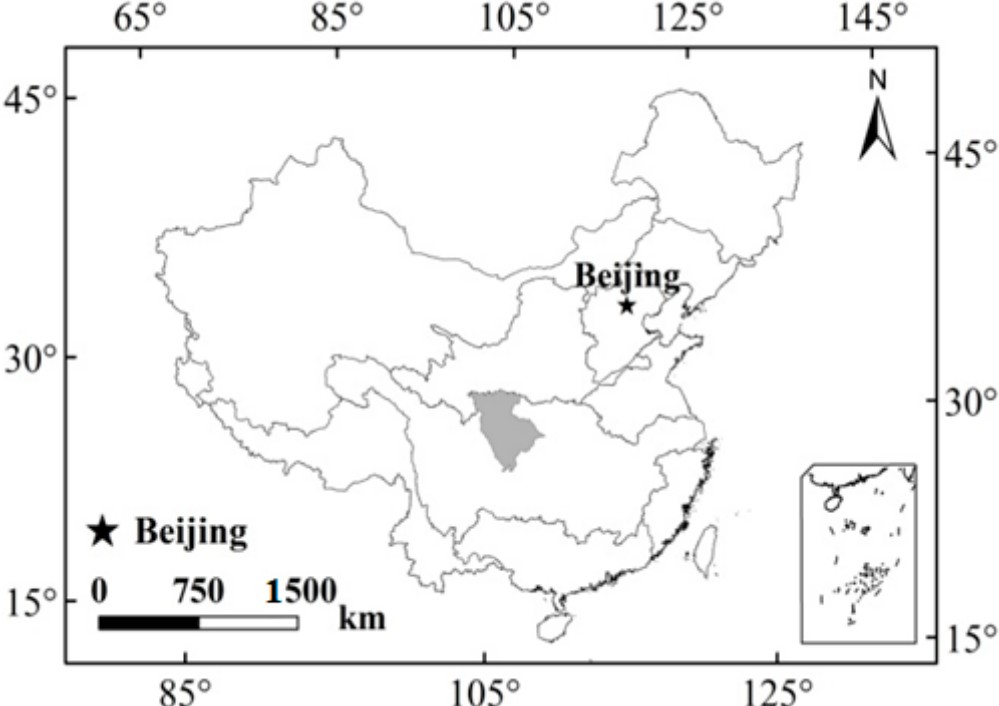

**Figure 1.** *Cont.*

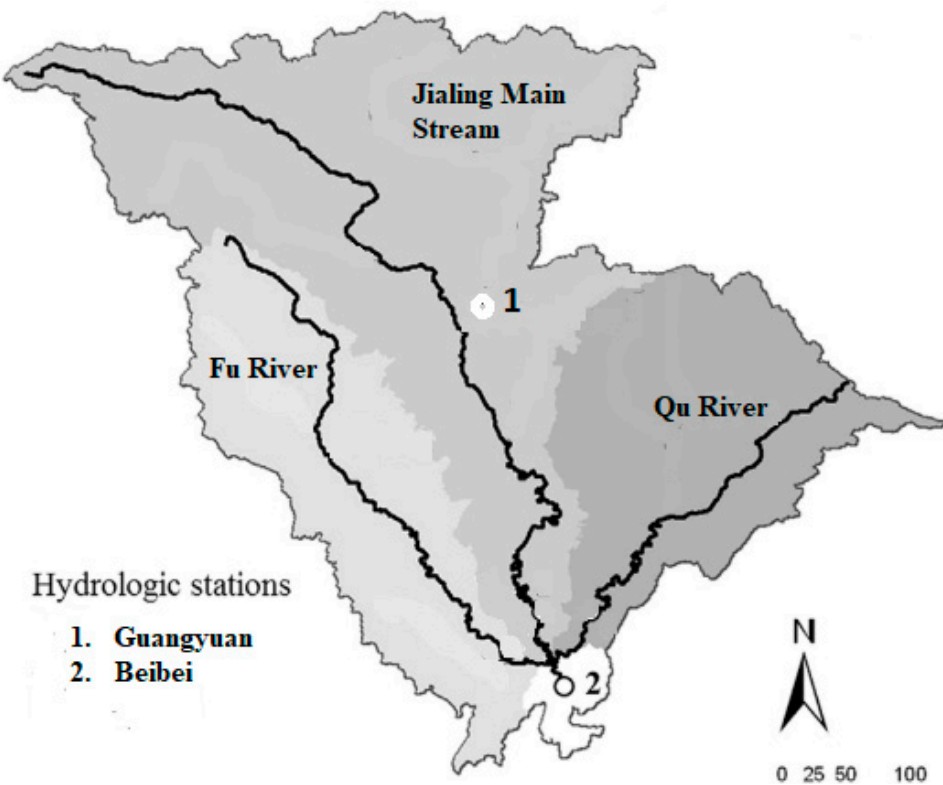

**Figure 1.** The geographical location of the study area.

*2.2. Adaptive Neuro-Fuzzy Inference System (ANFIS)*

Time series models are traditionally used to predict future phenomena in different fields of study. The major drawback of these models is their dependence on the autocorrelation factor. Machine learning models have been introduced and applied to overcome this foremost weakness of time series prediction models. ANFIS is one of the machine learning models that has been extensively used around the globe. It is a fusion of two different techniques, namely, artificial neural networks and fuzzy logic. Zadeh [48] developed fuzzy logic based on semantic uncertainty, which has been extensively used for modeling different environmental phenomena, including air pollution estimation and water pollution prediction. This fuzzy inference system (FIS) is pillared on three theories, i.e., the rule-based system, model's database, and inference system. The first part determines the if–then rules, the second part defines the membership function, and the third phase uses the rules to produce the results. The main problem of FIS is its incapability to optimize membership function and adjust parameters automatically. To overcome this problem, an artificial neural network (ANN) is coupled with FIS to develop ANFIS. The ANFIS model uses the training features of ANN to adjust the membership function of fuzzy logic. It is a multilayer feed-forward network based on ANN learning capabilities and fuzzy thinking. ANFIS has many membership functions, such as Gaussian, trapezoidal, sigmoid, generally bell-shaped, triangular, etc. In real-life problems, the selection of appropriate membership functions is vital. In the current study, the ANFIS model was developed using different input combinations and applied to sediment prediction at the study sites, Guangyuan and Beibei. The basic architecture of ANFIS is given in Figure 2.

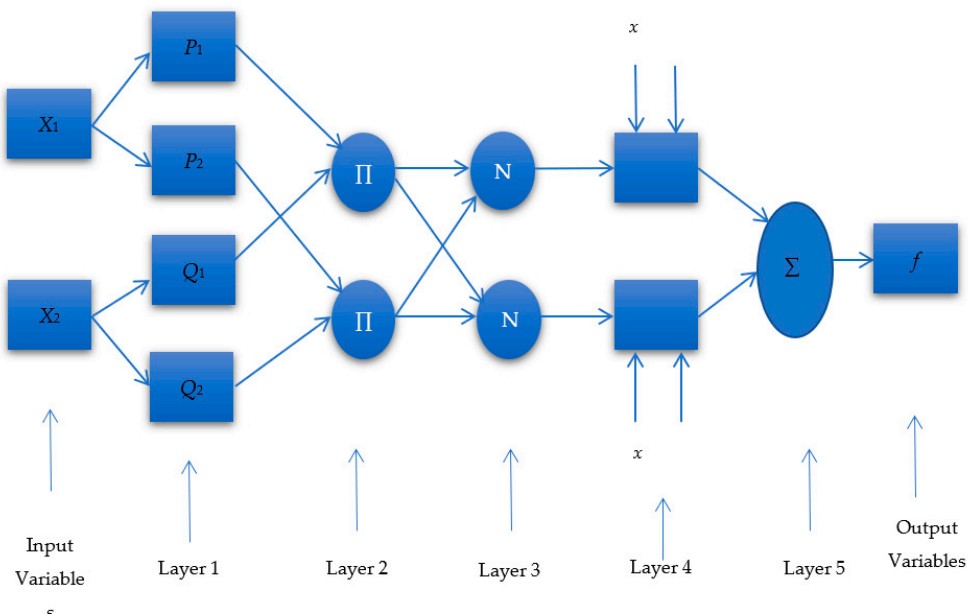

**Figure 2.** Basic architecture of ANFIS.

In Figure 2, $X_1$ and $X_2$ are inputs to the system, and $f$ is the output. The rules for the ANFIS system are described in Equations (1) and (2) below:

$$\text{Rule 1: IF } X_1 \text{ is } P_1 \text{ and } X_2 \text{ is } Q_1, \text{ then } f_1 = s_1 X_1 + t_1 X_2 + v_1 \tag{1}$$

$$\text{Rule 2: IF } X_1 \text{ is } P_2 \text{ and } X_2 \text{ is } Q_2, \text{ then } f_2 = s_2 X_1 + t_2 X_2 + v_2 \tag{2}$$

where $s_i$, $t_i$, and $v_i$ are linear output parameters that are required to optimize during model training. The working of all layers of ANFIS is discussed below [49,50].

Layer 1 (fuzzification layer): here, every node is considered an adaptive node, and each node passes the external signal to the next layer

$$O_{1,i} = \mu_{Pi}(X_1) \ for \ i = 1, \ 2, .. \tag{3}$$

$$O_{1,i} = \mu_{Qi-2}(X_2) \ for \ i = 1, \ 2, .. \tag{4}$$

In this step, the Gaussian membership function is used.

$$\mu_{Pi}(X_1, \sigma, c) = e^{\frac{-(X_1 - c)^2}{2\sigma^2}} \tag{5}$$

in which $\sigma$ and $c$ are membership function parameters. Therefore, the output of the first layer is as follows:

$$O_{1,i} = \mu_{Pi}(X_1) = e^{\frac{-(X_1 - c_i)^2}{2\sigma_i^2}} \tag{6}$$

Layer 2 (product layer): this layer produces the membership degree of inputs. All nodes of this layer are fixed nodes and labeled as $\prod$. The firing strength is calculated in this layer as

$$O_{2,i} = \mu_{Pi}(X_1)\mu_{Qi}(X_2) \ for \ i = 1, \ 2 \ldots \tag{7}$$

Layer 3 (normalized layer): here, the firing strength ratio at *i*-th rule to the sum of firing strengths of all rules is calculated,

$$O_{3,i} = \overline{w}_i = \frac{w_i}{w_1 + w_2} i = 1, \ 2, \ldots \tag{8}$$

Layer 4 (defuzzification layer): in this layer, each node is an adjustable node with the node function given below.

$$O_{4,i} = \overline{w}_i f_i = \overline{w}_i (s_i\, X_1 + t_i\, X_2 + v_i) \tag{9}$$

Here, $w_i$ is the output of the normalized firing strength, and $\{s_i, t_i, v_i\}$ are the parameters set of the node $i$.

Layer 5 (output layer): the final output of the model is calculated in this layer. It is the sum of incoming signals, as below.

$$O_{5,i} = \sum_i \overline{w}_i f_i = \frac{w_1 f_1 + w_2 f_2}{w_1 + w_2} \tag{10}$$

The ANFIS model is highly capable of learning and classifying input–output data.

### 2.3. M5 Tree Model

The M5 tree model was developed by Quinlan [51] and later rebuilt and enhanced by Wang and Witten [52]. This model is developed based on a decision tree concept, which links input and output variables. The model works in two steps. In the first step, input variables are divided into different groups based on linear regression. This minimizes the error of approximation between exact and forecast values. The standard deviation reduction plays an important role in fixing the division rule of the M5 tree model. Therefore, the decision tree depending upon input information is developed based on standard deviation reduction in the first step. In the second step, the tree is cropped from every leaf. Each further attempt involves finer classification levels, as these are further divided into branch and leaf nodes. The classification and regression tree (CART) algorithm is the basis of the M5 tree model. This model is trained based on the concept of standard deviation reduction (*SDR*), as discussed using the formula given below.

$$SDR = sd(T) - \sum \frac{|T_i|}{|T|} sd(T_i) \tag{11}$$

Here, *SDR* means standard deviation reduction; *sd* is expressed as a standard deviation; *T* represents a set of examples that reach the node; $T_i$ is the $i$th outcome of the possible set. The data's standard deviation (SD) is less than the parent nodes. The M5 tree model is developed in the current study to forecast SSL for seven different input combinations.

### 2.4. Multivariate Adaptive Regression Splines (MARS)

MARS, developed by Friedman [53], has undergone many reforms to enhance its performance. MARS's significant advantage is that it does not need any particular assumptions for mapping the input–output relationship, which is a major limitation of the M5 tree model. Therefore, the MARS model covers the limitation of the M5 tree model, where the endpoints of the segments are treated as nodes. MARS is the non-parametric regression model and is useful to forecast continuous numeric outcomes. The model's beauty is its flexible steps to manage relationships, which are almost additive or contain relations with other input variables to the model. MARS can explain the complex and nonlinear association between predictor and response variables. The MARS model can also work with the use of both the backward and forward stepwise procedure. Andres et al. [54] explain its use to remove preventable variables to improve the forecasting accuracy and make this model perform better during the backward stepwise procedure. The stepwise forward process helps to select the appropriate input variables for the MARS model.

Two basic functions with a range of inputs define the other variable. The variable $Y$, mapped from the variable $X$ with $c$ as the threshold, can be given below.

$$Y = max(0, X - c) \tag{12}$$

$$Y = max(0, c - X) \tag{13}$$

Here, two neighboring splines are used to have continuity in the basis function at the knot. The MARS model is widely used in many different fields, as it has properties to predict future approximation phenomena with good accuracy. In the present research, the MARS model is developed to forecast the sediment for seven different input combinations. The working procedure of MARS is shown in Figure 3.

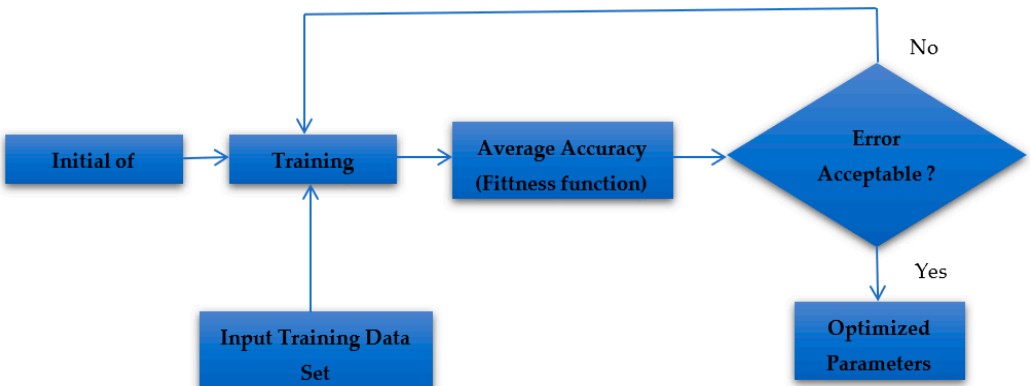

**Figure 3.** The working procedure of MARS.

### 2.5. K-Means (KM) Algorithm and MARS Hybrid Model

Lloyd [55] first introduced the KM technique in 1957 as a standard algorithm. Later, MacQueen [56] anticipated the term KM, a clustering technique based on the partition-based cluster analysis. This algorithm has been used in many different areas of research. The distance matrix used in the KM model is the Euclidean distance [57]. The first phase takes K initial seeds of clustering, and then the mean Euclidean distance is compared with each initial seed. This helps to assign the closest cluster seed. The process is iterated until the error is below the threshold. The choice of initial seed and the numbers of clusters are crucial, as they decide the KM method's accuracy [58]. The KM algorithm classifies the objects based upon characteristics into K number of groups.

If $J_i$ is the Euclidean distance, $x_k$ is the data vector, $c$ is the number of clusters, and $c_i$ is the cluster center, an objective function, $J$ is defined as

$$J = \sum_{i=1}^{c} J_i = \sum_{i=1}^{c} \left( \sum_{k, x_k \in G_i} \|x_k - c_i\|^2 \right) \tag{14}$$

In this study, the MARS model is coupled with the KM algorithm for better prediction results with the least error. The working procedure of MARS- KM is shown in Figure 4.

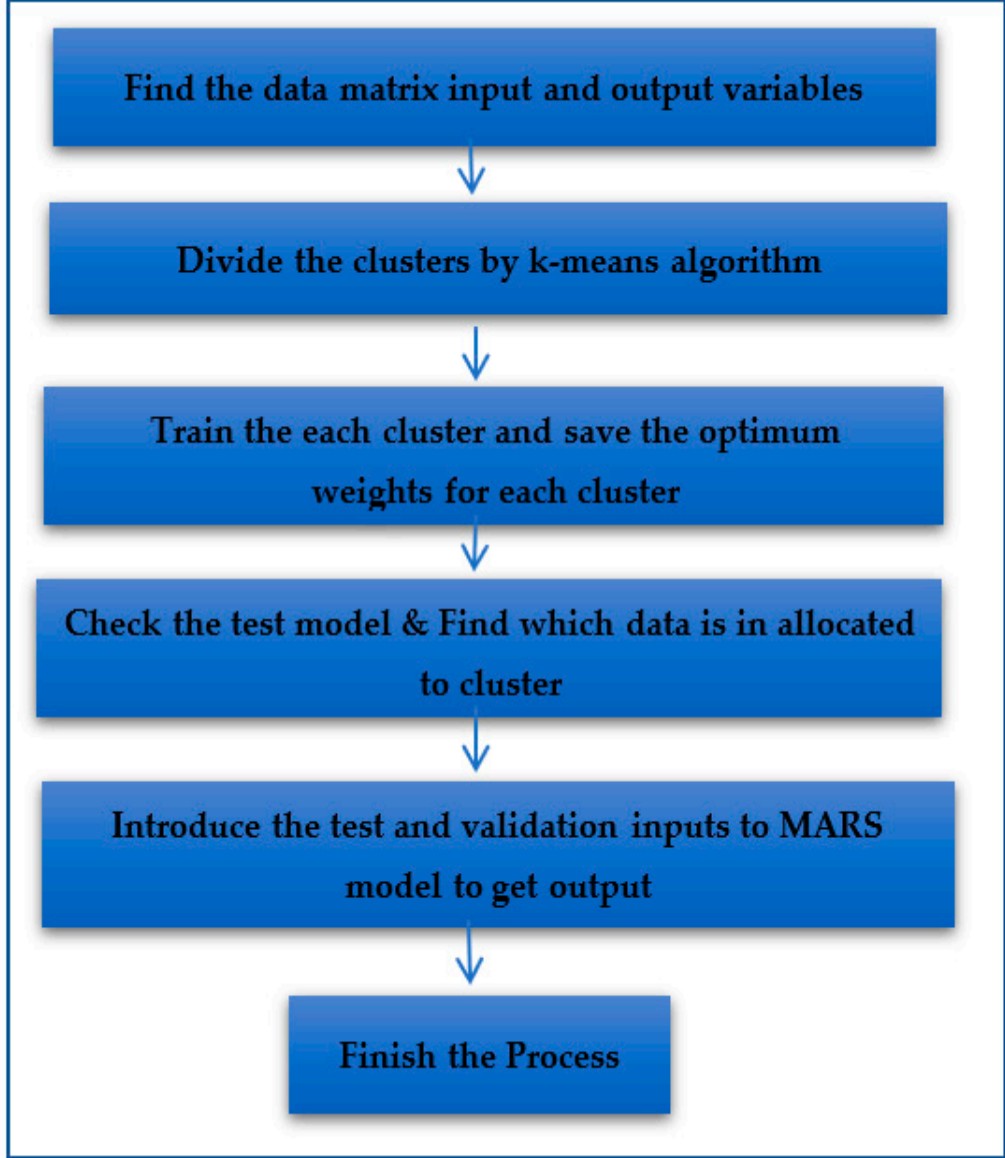

**Figure 4.** The working procedure of MARS–KMeans.

## 3. Application and Results

### 3.1. Modeling Approaches and Accuracy Assessments

Four approaches, adaptive neuro-fuzzy inference system (ANFIS), multivariate adaptive regression splines (MARS), M5 model tree (M5Tree), and MARS with k-means clustering algorithm (MARS–KM), were used for modeling suspended sediment using various input combinations of sediment load (St: kg/s) and streamflow (Q: $m^3$/s). The proposed models were developed using MATLAB software and compared using data at two stations, Guangyuan and Beibei. For each modeling approach, the sediment (kg/s) was modeled either separately using only the Q ($m^3$/s) measured at previous lags or combined with St (kg/s) estimated at previous lags. Here, the sediment (St: kg/s) is the response variable. The explanatory variables considered were varied from one to more inputs, formed by a combination of several Q and St lag values. In total, seven input combinations were compared, denoted as combinations (i), (ii) . . . etc. In the first four input combinations, only streamflow inputs were considered: (i) Qt; (ii) Qt and Qt-1; (iii) Qt, Qt-1, and Qt-2; and (iv) Qt, Qt-1, Qt-2, and Qt-3. After selecting the best Q-based input combination, after the fourth combination, sediment inputs were added to the best Q-based combination. For example, for the ANFIS method, the input combinations considered are (v) Qt and St-1;

(vi) Qt, St-1, and St-2, and (vi) Qt, St-1, St-2, and St-3, where Qt-1 and St-1 indicate the streamflow and sediment load at time t-1 (one previous day in this study). Performance assessment using different input combinations allows importance evaluation of variables and determining the lag values as inputs. Additionally, two different training scenarios were compared: splitting the dataset into two equal subsets having 50% of total data in each subset and a permutation between the two. Here, the two scenarios are denoted as the first training-test (scenario 1) and second training-test (scenario 2). In the first scenario, data from 4 January 2007, to 3 July 2011, were used for models' training, while the remaining data from 4 July 2011, to 31 December 2015, were used for model testing at both stations. In the second scenario, the training and test data sets were swapped (data from 4 July 2011, to 31 December 2015, were used for the models' training and data from 4 January 2007, to 3 July 2011, were used for the models' testing). These two scenarios allowed comparing the overall models' accuracy for the total range of the dataset. Model accuracy was computed by comparing the measured and the modeled data at each station separately using Nash–Sutcliffe efficiency (NSE), root mean squared error (RMSE: kg/s), and mean absolute error (MAE: kg/s).

*3.2. Comparison of Accuracy among Models: Guangyuan Station*

Table 1 shows the results obtained using ANFIS models for seven input combinations and two scenarios. In Table 1, Qt-1 and St-1 indicate the streamflow and sediment load at time t-1 (one previous day in this study) and vice versa. For only the *Q* as input, i.e., from input combination (i) Qt to input combination (iv) Qt, Qt-1, Qt-2, and Qt-3, the models showed a moderate to low accuracy, with mean NSE, RMSE, and MAE of 0.514, 1856.25 kg/s, and 346.75 kg/s, respectively.

**Table 1.** Performance of ANFIS model for different input combinations and training-test scenarios at Guangyuan Station.

| Statistics | Data Set | Input Combination | | | | | | |
|---|---|---|---|---|---|---|---|---|
| | | (i) | (ii) | (iii) | (iv) | (v) | (vi) | (vii) |
| RMSE | First training-test | 1649 | 1658 | 1725 | 1834 | 1470 | 1503 | 1608 |
| | Second training-test | 1677 | 1929 | 2003 | 2374 | 3162 | 3484 | 4496 |
| | Mean | 1663 | 1794 | 1864 | 2104 | 2316 | 2494 | 3052 |
| MAE | First training-test | 340 | 302 | 338 | 377 | 296 | 296 | 319 |
| | Second training-test | 323 | 350 | 355 | 387 | 608 | 565 | 209 |
| | Mean | 332 | 326 | 347 | 382 | 452 | 431 | 264 |
| NSE | First training-test | 0.563 | 0.559 | 0.522 | 0.46 | 0.653 | 0.637 | 0.585 |
| | Second training-test | 0.652 | 0.543 | 0.504 | 0.303 | 0.68 | 0.611 | 0.353 |
| | Mean | 0.608 | 0.551 | 0.513 | 0.382 | 0.667 | 0.624 | 0.469 |

In the table, input combinations are (i) Qt; (ii) Qt and Qt-1; (iii) Qt, Qt-1, and Qt-2; (iv) Qt, Qt-1, Qt-2, and Qt-3; (v) Qt and St-1; (vi) Qt, St-1, and St-2; and (vi) Qt, St-1, St-2, and St-3, where Qt-1 and St-1 indicate the streamflow and sediment load at time t-1 (one previous day in this study). Among the Q inputs, input combination (i) was considered, and therefore, after the fourth combination, sediment inputs were added to this combination.

The strong contribution of Qt is apparent, since there is no improvement in the models' performance after input combination (i). Instead, the mean NSE value dropped sharply from 0.608 to 0.382 (37.17%) after including time lags of Q as input in combinations (ii) to (iv). The mean RMSE also increased from 1663 to 2104 kg/s (20.96%) and the mean MAE from 326 to 382 kg/s or by 14.65%. This indicates that only Qt should be considered as a predictor for modeling St. The comparison of model performance for two scenarios, namely, the first training-test and the second training-test, showed little difference in the models' performance. The three statistical indices were relatively close to each other for the two scenarios. Table 1 shows a strong impact of sediment (St) on the ANFIS model accuracy. Beyond the input combination (iv), i.e., input combinations (v), (vi), and (vii), the ANFIS model showed a gradual performance improvement. The (St-1) combined with Qt (input combination (v)) provided the highest accuracy compared to the ANFIS model with only Qt as input (input combination (i)). The mean NSE increased by 8.85% for combination (v)

compared to input combination (i). However, it should be noted that the importance of St depends greatly on the number of lags included. Including more lags as predictors, i.e., the inclusion of two lags, St-1 and St-2 (input combination (vi)) or three lags, St-1, St-2, and St-3 (input combination (vi)), leads to a reduction in mean NSE by 6.44 and 29.68%, and an increase in mean RMSE by 24.11%, respectively. Overall, the best accuracy using the ANFIS model was achieved for input combination (v) with a mean NSE of 0.667.

The results obtained using the M5Tree model are reported in Table 2. The improvement achieved using the M5Tree model compared to the ANFS model was marginal. The M5Tree with the second input combination (Qt and Qt-1) yielded the best accuracy among the Q input-based models, with an average NSE value of 0.581. The prediction accuracy for the second input combination was higher than 1.20%, 11.87%, and 4.17% compared to that obtained using the input combinations of (i), (iii), and (iv), respectively. The difference in RMSE between the second and the fourth combination was the largest, an increase from 1716 to 1870 kg/s (8.23%.). To assess the impact of St on the model's performance, one to three St lags were combined with Qt and Qt-1 in input combinations (v) to (vii) (Table 2). The RMSE showed a slight decrease by ~1.87% and ~8.31% for input combination (iv) and (v) (models without St and with St), respectively.

**Table 2.** Performance of M5Tree model for different input combinations and training-test scenarios at Guangyuan Station.

| Statistics | Data Set | Input Combination | | | | | | |
|---|---|---|---|---|---|---|---|---|
| | | (i) | (ii) | (iii) | (iv) | (v) | (vi) | (vii) |
| RMSE | First training-test | 1756 | 1731 | 1963 | 1953 | 1622 | 1428 | 1428 |
| | Second training-test | 1700 | 1701 | 1702 | 1786 | 2047 | 1651 | 1651 |
| | Mean | 1728 | 1716 | 1833 | 1870 | 1835 | 1540 | 1540 |
| MAE | First training-test | 329 | 328 | 380 | 361 | 284 | 249 | 249 |
| | Second training-test | 311 | 310 | 314 | 334 | 304 | 269 | 269 |
| | Mean | 320 | 319 | 347 | 348 | 294 | 259 | 259 |
| NSE | First training-test | 0.505 | 0.519 | 0.381 | 0.394 | 0.577 | 0.672 | 0.672 |
| | Second training-test | 0.642 | 0.642 | 0.642 | 0.605 | 0.482 | 0.663 | 0.663 |
| | Mean | 0.574 | 0.581 | 0.512 | 0.500 | 0.530 | 0.668 | 0.668 |

In the table, input combinations are (i) Qt; (ii) Qt and Qt-1; (iii) Qt, Qt-1, and Qt-2; (iv) Qt, Qt-1, Qt-2, and Qt-3; (v) Qt, Qt-1, and St-1; (vi) Qt, Qt-1, St-1, and St-2; (vi) Qt, Qt-1, St-1, St-2, and St-3 where Qt-1 and St-1 indicate the streamflow and sediment load at time t-1 (one previous day in this study). Among the Q inputs, input combination (ii) was considered, and therefore, after the fourth combination, sediment inputs were added to this combination.

Nearly the same accuracy was achieved for all models using input combinations (vi) and (vii). The NSE was markedly higher and ranged from 0.663 to 0.672, with an average of 0.668 for those two input combinations. The accuracy increased most distinctly, with a decrease in RMSE and MAE by ~10.25 and ~18.80%, on average, between input combination (ii) and (vi). However, the inclusion of St-3 did not increase the NSE or decrease the RMSE and MAE. Consequently, M5Tree using the sixth and seventh combinations can be considered as the best models.

The statistical performance of the MARS model for both training scenarios is shown in Table 3. The results showed moderate MARS model accuracy for the first four input combinations (i, ii, iii, and iv), with a mean NSE value ranging from 0.567 to 0.595. It indicates a marginal gradual increase, yet more significant than that observed using ANFIS and M5Tree models. However, the improvement in the models' accuracy by increasing the number of inputs from one (Qt) to four (Qt, Qt-1, Qt-2, and Qt-3) was almost marginal, less than ~4.70% in NSE and ~3.37% and ~8.00%, in RMSE and MAE, respectively. Table 3 shows that the accuracy of the MARS models was higher for the second training-test dataset than the first training-test dataset for all input combinations. It means no substantial improvement with the increased number of inputs beyond two. Therefore, the combination (ii), having only Qt and Qt-1, was deemed for model development. The performances of MARS models improved significantly after the fourth input combination (Table 3). The mean NSE increased from 0.595 to 0.759 or by ~21.60%, the mean RMSE decreased from

1691 to 1312 kg/s or by ~22.41%, and the mean MAE dropped from 350 to 241 kg/s or by ~31.14%. The MARS model for the input combination (v) showed an overall higher accuracy than the sixth and seventh input combinations, having a slightly larger NSE value of 0.759. The MARS models' performance for the sixth and seventh combination was similar, with equal mean RMSE and MAE values of 1357 and 256 kg/s and a negligible mean NSE value of only 0.002. Overall, the MARS model for input combination (v) was the best model.

**Table 3.** Performance of MARS model for different input combinations and training-test scenarios at Guangyuan Station.

| Statistics | Data Set | Input Combination | | | | | | |
| --- | --- | --- | --- | --- | --- | --- | --- | --- |
| | | (i) | (ii) | (iii) | (iv) | (v) | (vi) | (vii) |
| RMSE | First training-test | 1717 | 1676 | 1676 | 1667 | 1225 | 1311 | 1311 |
| | Second training-test | 1783 | 1705 | 1715 | 1715 | 1399 | 1402 | 1402 |
| | Mean | 1750 | 1691 | 1696 | 1691 | 1312 | 1357 | 1357 |
| MAE | First training-test | 318 | 327 | 347 | 351 | 265 | 286 | 286 |
| | Second training-test | 325 | 312 | 349 | 349 | 217 | 226 | 226 |
| | Mean | 322 | 320 | 348 | 350 | 241 | 256 | 256 |
| NSE | First training-test | 0.526 | 0.549 | 0.549 | 0.554 | 0.759 | 0.72 | 0.724 |
| | Second training-test | 0.607 | 0.64 | 0.636 | 0.636 | 0.758 | 0.757 | 0.757 |
| | Mean | 0.567 | 0.595 | 0.593 | 0.595 | 0.759 | 0.739 | 0.741 |

In the table, for the first training-test data set, the input combinations are (i) $Q_t$; (ii) $Q_t$ and $Q_{t-1}$; (iii) $Q_t$, $Q_{t-1}$, and $Q_{t-2}$; (iv) $Q_t$, $Q_{t-1}$, $Q_{t-2}$, and $Q_{t-3}$; (v) $Q_t$, $Q_{t-1}$, $Q_{t-2}$, $Q_{t-3}$, and $S_{t-1}$; (vi) $Q_t$, $Q_{t-1}$, $Q_{t-2}$, $Q_{t-3}$, $S_{t-1}$, and $S_{t-2}$; and (vi) $Q_t$, $Q_{t-1}$, $Q_{t-2}$, $Q_{t-3}$, $S_{t-1}$, $S_{t-2}$, and $S_{t-3}$, where $Q_{t-1}$ and $S_{t-1}$ indicate the streamflow and sediment load at time t-1 (one previous day in this study). Among the Q inputs, input combination (vi) was considered, and therefore, after the fourth combination, sediment inputs were added to this combination. Similarly, for the second training-test data set, the input combinations are (i) $Q_t$; (ii) $Q_t$ and $Q_{t-1}$; (iii) $Q_t$, $Q_{t-1}$, and $Q_{t-2}$; (iv) $Q_t$, $Q_{t-1}$, $Q_{t-2}$, and $Q_{t-3}$; (v) $Q_t$, $Q_{t-1}$, and $S_{t-1}$; (vi) $Q_t$, $Q_{t-1}$, $S_{t-1}$, and $S_{t-2}$; and (vi) $Q_t$, $Q_{t-1}$, $S_{t-1}$, $S_{t-2}$, and $S_{t-3}$.

The statistical performance of the MARS–KM model for different input combinations and training scenarios is given in Table 4. Good accuracy using MARS–KM was observed for all the input combinations in terms of all three statistical metrics. The performance was higher compared to the ANFIS, M5Tree, and MARS models. The mean RMSE was higher (1342 kg/s) for the first four input combinations ((i) to (iv)) than for the last four input combinations ((v) to (vii)) (1158 kg/s). The differences observed for the four first input combinations were as follows: (1) good prediction accuracy using the MARS–KM model for the fourth input combination ($Q_t$, $Q_{t-1}$, $Q_{t-2}$, and $Q_{t-3}$) with a mean NSE value of 0.813, a mean RMSE value of 1158 kg/s, and a mean MAE value of 210 kg/s; (2) a slight to significant difference of mean RMSE, equal to 3.105% and 13.711% between the fourth and first combinations, and second and third combinations, respectively; and (3) the MAE of MARS–KM models' accuracy dropped significantly for the fourth input combination. The inclusion of different lags of $S_t$ as input showed a marked improvement in the models' accuracy. For the three last input combinations (v, vi, and vii), the mean RMSE and MAE values rapidly decreased (dropped from 1158 to 1144 kg/s), while the NSE slightly increased from 0.813 to 0.818. Overall, MARS–KM also showed better performance for the fifth input combination.

The comparison of four machine learning methods with the corresponding best input combination is shown in Table S2. The four models exhibited different accuracy varying with mean NSE ranging from 0.608 to 0.818, mean RMSE between 1143 and 1663 kg/s, and mean MAE between 177 and 332 kg/s. The MARS–KM enhanced the $S_t$ prediction significantly, while the ANFIS showed the least accuracy compared to the other models. The results highlight that, although the models were developed using the same input variables, their capabilities in capturing the major variability and uncertainty in the dataset were variable. The most apparent difference between the models was between MARS–KM and the ANFIS. The MARS–KM improved accuracy compared to ANRIS by 21%, 31.27%, and 44.53% in terms of NSE, RMSE, and MAE, respectively. The M5Tree and MARS models lie between the two extremes. The differences between the two showed that the MARS algorithm generally enhanced the M5Tree accuracy by an average of 14.80% and

6.95% reduction of RMSE and MAE, respectively. Table S2 indicates that, on average, the results obtained using different models differ significantly in terms of different metrics. For example, the variation of RMSE was lower than 12% between MARS and MARS–KM and 31% between ANFIS and MARS–KM. Overall, the models can be ranked in decreasing performance order as MARS–KM, MARS, M5Tree, and ANFIS.

**Table 4.** Performance of MARS–KM model for different input combinations and training-test scenarios at Guangyuan Station.

| Statistics | Data Set | Input Combination | | | | | | |
|---|---|---|---|---|---|---|---|---|
| | | (i) | (ii) | (iii) | (iv) | (v) | (vi) | (vii) |
| RMSE | First training-test | 1103 | 1400 | 1108 | 1029 | 1003 | 1003 | 1002 |
| | Second training-test | 1285 | 1284 | 1284 | 1286 | 1282 | 1284 | 1285 |
| | Mean | 1194 | 1342 | 1196 | 1158 | 1143 | 1144 | 1144 |
| MAE | First training-test | 216 | 280 | 223 | 232 | 187 | 187 | 185 |
| | Second training-test | 188 | 185 | 185 | 187 | 167 | 168 | 170 |
| | Mean | 202 | 233 | 204 | 210 | 177 | 178 | 178 |
| NSE | First training-test | 0.805 | 0.685 | 0.803 | 0.83 | 0.838 | 0.838 | 0.839 |
| | Second training-test | 0.796 | 0.796 | 0.796 | 0.795 | 0.797 | 0.796 | 0.796 |
| | Mean | 0.801 | 0.741 | 0.800 | 0.813 | 0.818 | 0.817 | 0.818 |

In the table, for the first training-test data set, the input combinations are: (i) $Q_t$; (ii) $Q_t$ and $Q_{t-1}$; (iii) $Q_t$, $Q_{t-1}$, and $Q_{t-2}$; (iv) $Q_t$, $Q_{t-1}$, $Q_{t-2}$, and $Q_{t-3}$; (v) $Q_t$, $Q_{t-1}$, $Q_{t-2}$, $Q_{t-3}$, and $S_{t-1}$; (vi) $Q_t$, $Q_{t-1}$, $Q_{t-2}$, $Q_{t-3}$, $S_{t-1}$, and $S_{t-2}$; and (vi) $Q_t$, $Q_{t-1}$, $Q_{t-2}$, $Q_{t-3}$, $S_{t-1}$, $S_{t-2}$, and $S_{t-3}$, where $Q_{t-1}$ and $S_{t-1}$ indicate the streamflow and sediment load at time t-1 (one previous day in this study). Among the Q inputs, input combination (vi) was considered, and therefore, after the fourth combination, sediment inputs were added to this combination. Similarly, for the second training-test data set, the input combinations are (i) $Q_t$; (ii) $Q_t$ and $Q_{t-1}$; (iii) $Q_t$, $Q_{t-1}$, and $Q_{t-2}$; (iv) $Q_t$, $Q_{t-1}$, $Q_{t-2}$, and $Q_{t-3}$; (v) $Q_t$ and $S_{t-1}$; (vi) $Q_t$, $S_{t-1}$, and $S_{t-2}$; and (vi) $Q_t$, $S_{t-1}$, $S_{t-2}$, and $S_{t-3}$.

### 3.3. Comparison of Accuracy among Models: Beibei Station

The results obtained using the ANFIS model for all input combinations and training scenarios at the Beibei Station are presented in Table 5. The results showed ANFIS models for the first four input combinations, (i) $Q_t$; (ii) $Q_t$ and $Q_{t-1}$; (iii) $Q_t$, $Q_{t-1}$, and $Q_{t-2}$; and (iv) $Q_t$, $Q_{t-1}$, $Q_{t-2}$, and $Q_{t-3}$, yielded relatively similar mean RMSE and MAE, ranging from 3553 to 3628 kg/s and 610 to 710 kg/s, respectively, whereas the fourth input combination showed the highest mean RMSE and lowest NSE.

**Table 5.** Performance of ANFIS model for different input combinations and training-test scenarios at Beibei Station.

| Statistics | Data Set | Input Combination | | | | | | |
|---|---|---|---|---|---|---|---|---|
| | | (i) | (ii) | (iii) | (iv) | (v) | (vi) | (vii) |
| RMSE | First training-test | 4003 | 3982 | 4064 | 4087 | 3591 | 3441 | 3909 |
| | Second training-test | 3163 | 3123 | 3159 | 3168 | 2668 | 3295 | 3254 |
| | Mean | 3583 | 3553 | 3612 | 3628 | 3130 | 3368 | 3582 |
| MAE | First training-test | 743 | 644 | 663 | 673 | 715 | 746 | 796 |
| | Second training-test | 677 | 576 | 643 | 632 | 474 | 511 | 529 |
| | Mean | 710 | 610 | 653 | 653 | 595 | 629 | 663 |
| NSE | First training-test | 0.52 | 0.525 | 0.505 | 0.499 | 0.614 | 0.645 | 0.546 |
| | Second training-test | 0.68 | 0.688 | 0.681 | 0.679 | 0.772 | 0.653 | 0.661 |
| | Mean | 0.600 | 0.607 | 0.593 | 0.589 | 0.693 | 0.649 | 0.602 |

In the table, input combinations are (i) $Q_t$; (ii) $Q_t$ and $Q_{t-1}$; (iii) $Q_t$, $Q_{t-1}$, and $Q_{t-2}$; (iv) $Q_t$, $Q_{t-1}$, $Q_{t-2}$, and $Q_{t-3}$; (v) $Q_t$ and $S_{t-1}$; (vi) $Q_t$, $S_{t-1}$, and $S_{t-2}$; and (vi) $Q_t$, $S_{t-1}$, $S_{t-2}$, and $S_{t-3}$, where $Q_{t-1}$ and $S_{t-1}$ indicate the streamflow and sediment load at time t-1 (one previous day in this study). Among the Q inputs, input combination (i) was considered, and therefore, after the fourth combination, sediment inputs were added to this combination. Similarly, for the second training-test data set, the input combinations are (i) $Q_t$, (ii) $Q_t$ and $Q_{t-1}$; (iii) $Q_t$, $Q_{t-1}$, and $Q_{t-2}$; (iv) $Q_t$, $Q_{t-1}$, $Q_{t-2}$, and $Q_{t-3}$; (v) $Q_t$, $Q_{t-1}$, and $S_{t-1}$; (vi) $Q_t$, $Q_{t-1}$, $S_{t-1}$, and $S_{t-2}$; and (vi) $Q_t$, $Q_{t-1}$, $S_{t-1}$, $S_{t-2}$, and $S_{t-3}$.

Therefore, only $Q_t$ was combined with different lags of $S$ to form the input combinations of (v), (vi), and (vii). The results showed a strong to moderate improvement in accuracy, with mean NSE value ranging from 0.602 to 0.693, mean RMSE ranging from 3130 to 3582 kg/s, and a mean MAE between 595 and 663 kg/s. The highest mean NSE value of 0.693 was found for the input combination (v). Relatively low mean NSE of 0.602 and

large mean RMSE and MAE were found for the input combination (vii), highlighting the negligible contribution of St-2 and St-3 (Table 5). The increasing number of input variables showed a minimal contribution to ANFIS model performance improvement at this station.

Results obtained for the M5Tree model are reported in Table 6. Suspended sediment simulated from the first four input combinations showed a weak and insignificant difference between the models, with a slight superiority of the first input combination (only the Qt). Table 6 shows the variation of the RMSE, MAE, and NSE values averaged for different input combinations.

**Table 6.** Performance of M5Tree model for different input combinations and training-test scenarios at Beibei Station.

| Statistics | Data Set | Input Combination | | | | | | |
|---|---|---|---|---|---|---|---|---|
| | | (i) | (ii) | (iii) | (iv) | (v) | (vi) | (vii) |
| RMSE | First training-test | 4567 | 4771 | 4596 | 4728 | 4426 | 3221 | 3221 |
| | Second training-test | 3217 | 3218 | 3951 | 3890 | 3301 | 3019 | 2953 |
| | Mean | 3892 | 3995 | 4274 | 4309 | 3864 | 3120 | 3087 |
| MAE | First training-test | 733 | 782 | 726 | 794 | 680 | 495 | 495 |
| | Second training-test | 566 | 569 | 638 | 612 | 494 | 464 | 451 |
| | Mean | 650 | 676 | 682 | 703 | 587 | 480 | 473 |
| NSE | First training-test | 0.375 | 0.318 | 0.367 | 0.33 | 0.413 | 0.689 | 0.689 |
| | Second training-test | 0.669 | 0.668 | 0.5 | 0.516 | 0.651 | 0.708 | 0.721 |
| | Mean | 0.522 | 0.493 | 0.434 | 0.423 | 0.532 | 0.699 | 0.705 |

In the table, input combinations are (i) Qt; (ii) Qt and Qt-1; (iii) Qt, Qt-1, and Qt-2; (iv) Qt, Qt-1, Qt-2, and Qt-3; (v) Qt and St-1; (vi) Qt, St-1, and St-2; and (vi) Qt, St-1, St-2, and St-3, where Qt-1 and St-1 indicate the streamflow and sediment load at time t-1 (one previous day in this study). Among the Q inputs, input combination (i) was considered, and therefore, after the fourth combination, sediment inputs were added to this combination.

The results showed an apparent decrease in the model's performances from the input combination (i) to the input combination (iv). The mean RMSE and MAE values increased from 3892 to 4309 kg/s (9.67%) and from 566 to 612 kg/s (7.51%), respectively, and the mean NSE value gradually declined to its lowest value of 0.423. This negligible difference in the models' accuracy might be due to the marginal effect of the higher lag streamflow data, which have already been highlighted in the previous discussion. Therefore, the inclusion of only Qt was sufficient for predicting suspended sediment. The effect of an increasing number of input variables on M5Tree performances can also be observed in Table 6. There was a positive effect of St on model accuracy. Interestingly, the suspended sediment was more sensitive to its antecedent values than the Q. The lowest RMSE and MAE values of 3087 and 473 kg/s, respectively, and the mean NSE of 0.705 were obtained using the input combination (vii) (Qt, St-1, St-2, and St-3). The significant increase in the NSE value from 0.522 to 0.705 (18.3%) indicates that the inclusion of St-1, St-2, and St-3 has significantly contributed to M5Tree performances. Nevertheless, the sensitivity of the M5Tree model to the inclusion of the St was not the same for both training-test scenarios. The improvement was more significant for the second training-test data set.

The performances of the MARS model for different input combinations are shown in Table 7. An overall maximum mean NSE value of 0.545 was obtained using only the Qt as the input (input combination (i)), suggesting a weak level of agreement between measured and predicted suspended sediment. Mean RMSE and MAE values of 3822 and 631 kg/s were achieved using only the Qt, and the level of accuracy remained very low regardless of the number of Q lags included from one to four. Table 7 revealed Qt as the most dominant input variable. Therefore, only the first combination was coupled with different lag values of sediment. The St simulations for the input combinations (v), (vi), and (vii) showed better performance (Table 7). The NSE ranged from 0.706 to 0.728, with a mean of 0.720. The performances of the input combination (vi) (Qt, St-1, and St-2) and input combination (vii) (Qt, St-1, St-2, and St-3) were quite similar, where the input combination (v) performed less well with a lower mean NSE (0.706) and higher RMSE (3070 kg/s) and MAE (534 kg/s)

values. The above results indicate that the input combination (vii) should be used to obtain the best accuracy.

**Table 7.** Performance of MARS model for different input combinations and training-test scenarios at Beibei Station.

| Statistics | Data Set | Input Combination | | | | | | |
|---|---|---|---|---|---|---|---|---|
| | | (i) | (ii) | (iii) | (iv) | (v) | (vi) | (vii) |
| RMSE | First training-test | 4250 | 4451 | 4431 | 4355 | 3453 | 3321 | 3265 |
| | Second training-test | 3394 | 3344 | 3375 | 3296 | 2686 | 2616 | 2650 |
| | Mean | 3822 | 3898 | 3903 | 3826 | 3070 | 2969 | 2958 |
| MAE | First training-test | 702 | 763 | 766 | 826 | 561 | 581 | 575 |
| | Second training-test | 560 | 602 | 589 | 643 | 507 | 483 | 462 |
| | Mean | 631 | 683 | 678 | 735 | 534 | 532 | 519 |
| NSE | First training-test | 0.459 | 0.406 | 0.412 | 0.432 | 0.643 | 0.67 | 0.681 |
| | Second training-test | 0.631 | 0.642 | 0.635 | 0.652 | 0.769 | 0.781 | 0.775 |
| | Mean | 0.545 | 0.524 | 0.524 | 0.542 | 0.706 | 0.726 | 0.728 |

In the table, for the first training-test data set, the input combinations are (i) Qt; (ii) Qt and Qt-1; (iii) Qt, Qt-1, and Qt-2; (iv) Qt, Qt-1, Qt-2, and Qt-3; (v) Qt and St-1; (vi) Qt, St-1, and St-2; and (vi) Qt, St-1, St-2, and St-3, where Qt-1 and St-1 indicate the streamflow and sediment load at time t-1 (one previous day in this study). Among the Q inputs, input combination (i) was considered, and therefore, after the fourth combination, sediment inputs were added to this combination. Similarly, for the second training-test data set, the input combinations are (i) Qt; (ii) Qt and Qt-1; (iii) Qt, Qt-1, and Qt-2; (iv) Qt, Qt-1, Qt-2, and Qt-3; (v) Qt, Qt-1, Qt-2, Qt-3, and St-1; (vi) Qt, Qt-1, Qt-2, Qt-3, St-1, and St-2; and (vi) Qt, Qt-1, Qt-2, Qt-3, St-1, St-2, and St-3.

Table 8 summarizes the statistics of MARS–KM models. MARS–KM at Beibei Station showed exceptional performance compared to other models. The results indicated the model's higher ability to predict the suspended sediment independently and successfully, even without the inclusion of St lags as input. The NSE values for the seven input combinations were in the range of 0.810 to 8.17 with a mean of ~0.754. The best accuracy was achieved using only the Qt and Qt-1 as inputs (input combination (ii)), with a mean NSE of 0.817. The RMSE and MAE for the best model (input combination (ii)) were much smaller, ~2428 and ~2428 kg/s, respectively. The results described above indicate the MARS–KM model's effectiveness in predicting suspended sediment.

**Table 8.** Performance of MARS–KM model for different input combinations and training-test scenarios at Beibei Station.

| Statistics | Data Set | Input Combination | | | | | | |
|---|---|---|---|---|---|---|---|---|
| | | (i) | (ii) | (iii) | (iv) | (v) | (vi) | (vii) |
| RMSE | First training-test | 2664 | 2377 | 2664 | 2664 | 2258 | 2419 | 2403 |
| | Second training-test | 2950 | 2478 | 2914 | 2918 | 2921 | 2534 | 2525 |
| | Mean | 2807 | 2428 | 2789 | 2791 | 2590 | 2477 | 2464 |
| MAE | First training-test | 508 | 434 | 518 | 524 | 382 | 337 | 334 |
| | Second training-test | 429 | 367 | 421 | 417 | 485 | 458 | 471 |
| | Mean | 469 | 401 | 470 | 471 | 434 | 398 | 403 |
| NSE | First training-test | 0.787 | 0.831 | 0.787 | 0.787 | 0.847 | 0.825 | 0.827 |
| | Second training-test | 0.721 | 0.803 | 0.728 | 0.728 | 0.727 | 0.794 | 0.796 |
| | Mean | 0.754 | 0.817 | 0.758 | 0.758 | 0.787 | 0.810 | 0.812 |

In the table, for the first training-test data set, the input combinations are (i) Qt; (ii) Qt and Qt-1; (iii) Qt, Qt-1, and Qt-2; (iv) Qt, Qt-1, Qt-2, and Qt-3; (v) Qt, Qt-1, and St-1; (vi) Qt, Qt-1, St-1, and St-2; and (vi) Qt, Qt-1, St-1, St-2, and St-3, where Qt-1 and St-1 indicate the streamflow and sediment load at time t-1 (one previous day in this study). Among the Q inputs, input combination (vi) was considered, and therefore, after the fourth combination, sediment inputs were added to this combination. Similarly, for the second training-test data set, the input combinations are (i) Qt; (ii) Qt and Qt-1; (iii) Qt, Qt-1, and Qt-2; (iv) Qt, Qt-1, Qt-2, and Qt-3; (v) Qt, Qt-1, Qt-2, Qt-3, and St-1; (vi) Qt, Qt-1, Qt-2, Qt-3, St-1, and St-2; and (vi) Qt, Qt-1, Qt-2, Qt-3, St-1, St-2, and St-3.

The relative performance of the four models with the best input combinations is presented in Table S3. The results demonstrated that the performances of the models were relatively far from excellent. None of the models, i.e., ANFIS, M5Tree, MARS, and MARS–KM, achieved an NSE higher than 0.90. Overall, the performances of MARS–KM were remarkably superior. The MARS–KM improved the mean NSE of ANFIS, M5Tree, and MARS by 12.4%, 11.2%, and 8.9%, respectively. Additionally, it reduced the RMSE of

ANFIS, M5Tree, and MARS by 22.42%, 21.34%, and 19.91%, respectively, and the MAE by 32.60%, 15.22%, and 22.73%, respectively.

The suspended sediment estimation ability of machine learning models was further assessed through visual comparison with the in-situ data (Figure 5a–d). MARS–KM and MARS reproduced suspended sediment variation much better and thus enhanced M5Tree and ANFIS models' prediction accuracy. The scatterplots (Figures S2–S5, see Supplementary Materials) revealed that (i) the underestimated data points were much higher than the overestimated data points, and (ii) all the models failed to simulate large suspended sediment values. It is apparent from Figure 6 that the MARS–KM shows superiority in simulating cumulative sediment amounts compared to other alternatives.

In a recently published paper, Juez et al. [59] studied the sediment hysteresis, a direct link among sediment size, distal sediment supply, and proximal sediment data obtained through a laboratory experiment. The authors demonstrated that sediment in the channel downstream depends mainly on the time-varying sediment load with different hysteresis types. The study also highlighted that sediment availability is governed by the evolution of two important morphological parts of the riverbed, degradation and aggradation. The shapes of hysteresis loops have often been intrinsically correlated to these two morphological processes. Finally, one of the important findings of the above-reported investigation is that the sediment concentration–discharge hysterical behavior is increasingly likely to amount between the distal sediment supply and the proximal sediment availability.

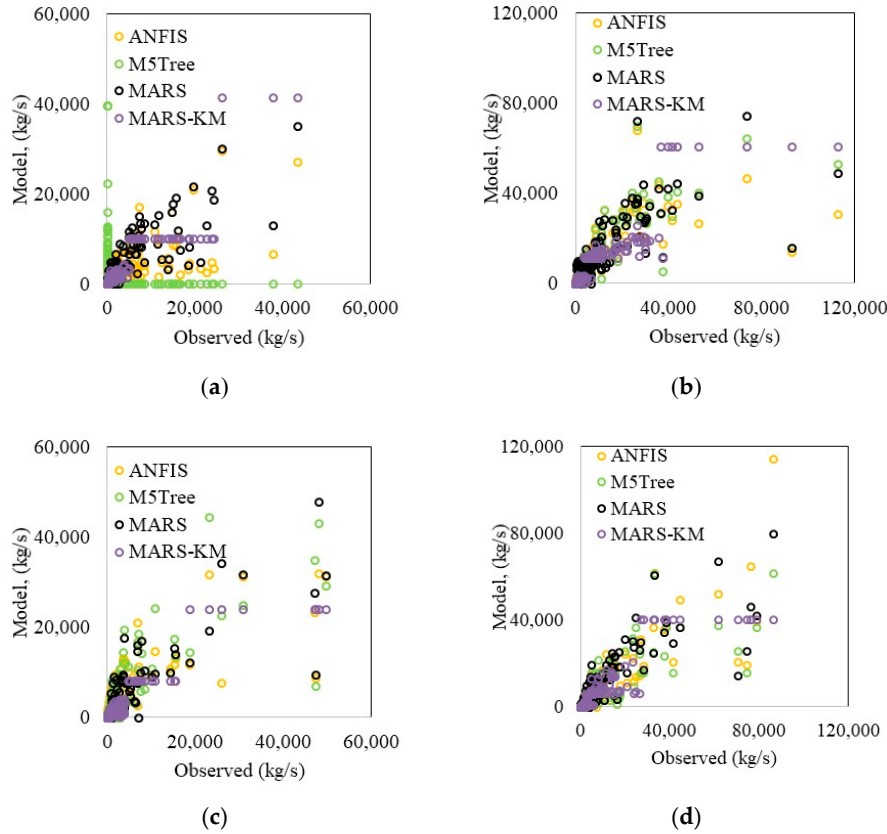

**Figure 5.** Scatterplots of the observed and estimated sediments by ANFIS, M5Tree, MARS, and MARS–KM models in the test period at (**a**) Guangyuan Station for the first training-test scenario, (**b**) Guangyuan Station for the second training-test scenario, (**c**) Beibei Station for the first training-test scenario, and (**d**) Beibei Station for the second training-test scenario.

Suspended sediment concentration (*SSC*) and discharge (*Q*) are two variables with a temporal shift; consequently, it is important to focus on these approaches' weaknesses

and limitations in modeling these kinds of variables. Based on the direct linking of SSC to Q, the empirical models can estimate SSC reasonably. However, data-driven models resulted in slightly better performances in predicting the amount of SSC. Interpretation of the empirical models is more straightforward, as physical, morphological, and hydrological processes are explicitly expressed with simplified equations. However, it is appropriate for data-driven black-box models to examine whether it is possible to simulate SSC with input values outside the data range used during their calibration. Indeed, both the empirical and data-driven models showed encouraging results in terms of model accuracy; physical interpretation of the data-driven models is a challenge, thus requiring further analysis in quantifying the correlation between the input and output variables in a more meaningful way. When extending the models to another watershed, the empirical models may be more practical because of local calibration. Additionally, most input variables (i.e., the Q) undergo a rapid momentary fluctuation, which is very hard to capture with data-driven models. This deficiency makes the data-driven models unable to generate a durable and continuous response. Clearly, although the advantages exist, the limitations are always existing, which should be addressed in future studies.

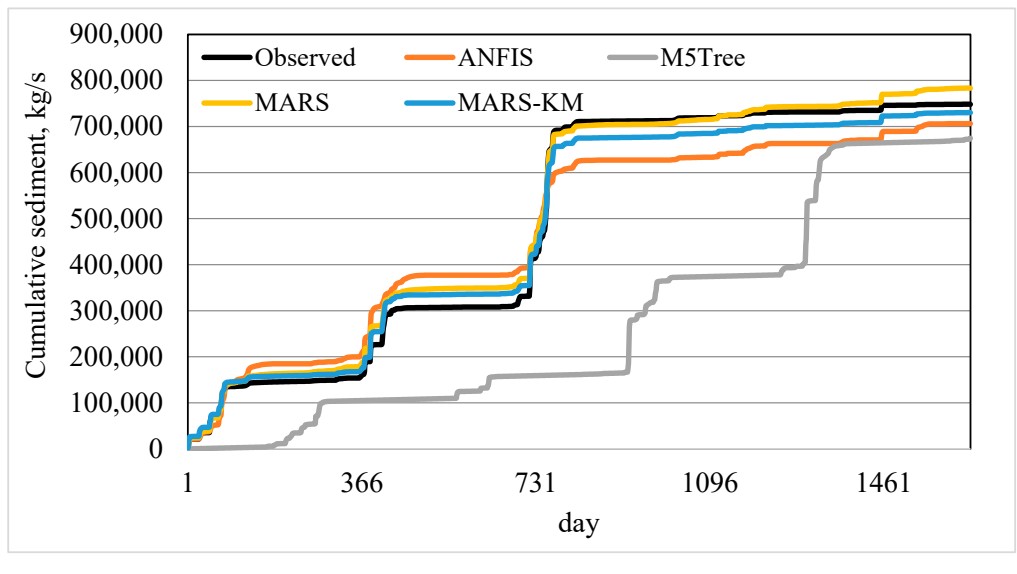

(**a**)

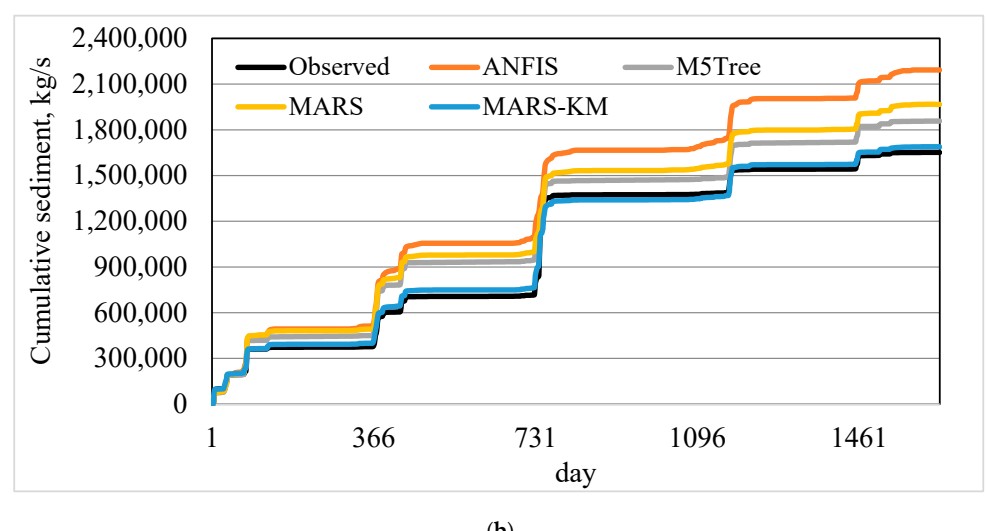

(**b**)

**Figure 6.** *Cont.*

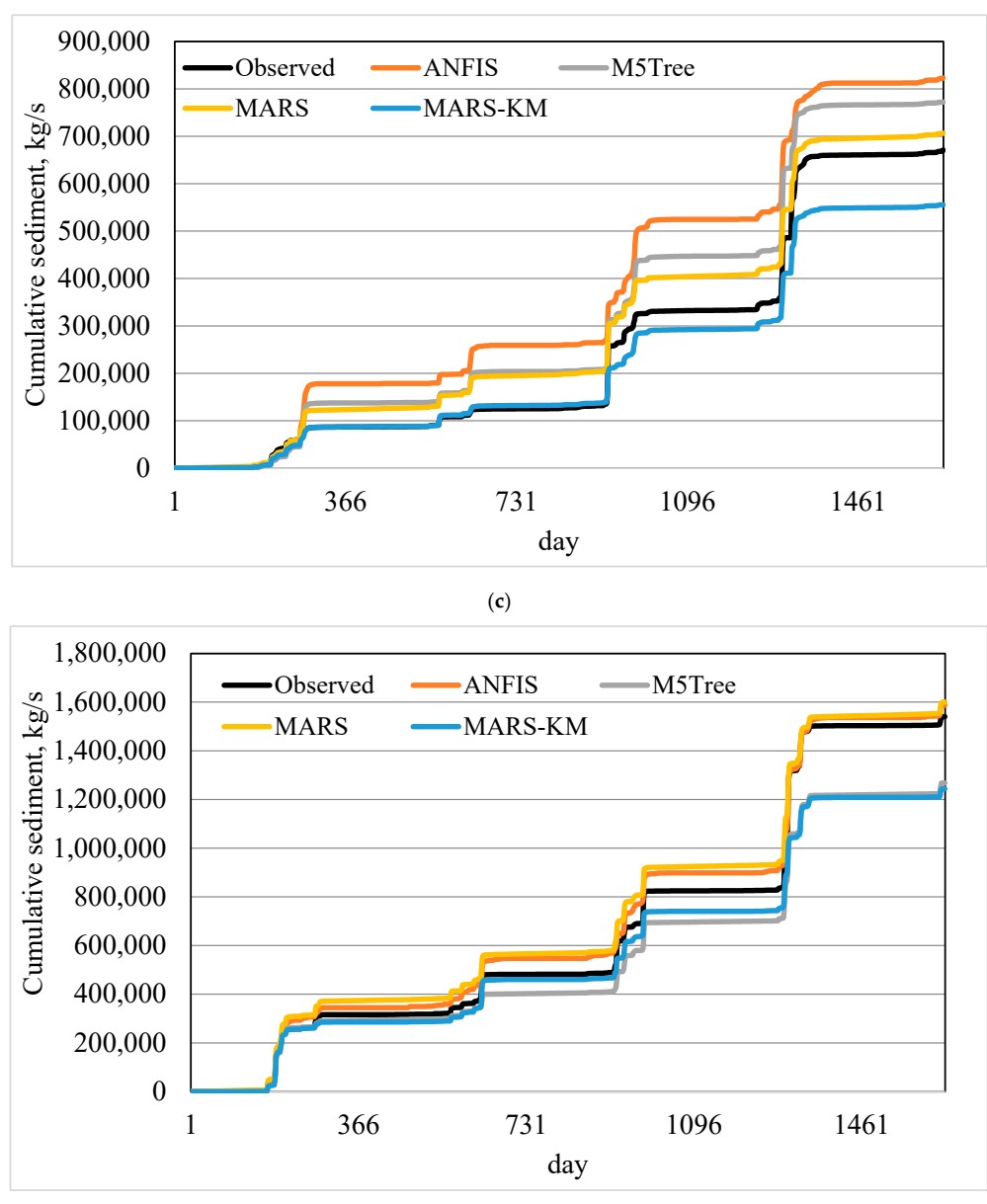

(**c**)

(**d**)

**Figure 6.** Cumulative sediment amounts produced by the ANFIS, M5Tree, MARS, and MARS–KM models in the test period: (**a**) Guangyuan Station for the first training-test scenario, (**b**) Guangyuan Station for the second training-test scenario, (**c**) Beibei Station for the first training-test scenario, and (**d**) Beibei Station for the first training-test scenario.

## 4. Conclusions

In this investigation, a new method was developed by hybridizing MARS and the K-means clustering algorithm to improve the accuracy of suspended sediment prediction. The models were developed using daily discharge and sediment data at two stations in China. The MARS–KM models' performance was compared with ANFIS, MARS, and M5Tree models using three statistical metrics, RMSE, MAE, and NSE, and graphical comparison. The following conclusions were reached from the outcomes of the presented work:

The proposed MARS–KM considerably improved the accuracy of the ANFIS, MARS, and M5Tree methods. The increments in the RMSE of the three mentioned methods were by 39%, 30%, and 18%, and 24%, 22%, and 8% for the first and second scenarios at the Guangyuan Station, and by 34%, 26%, and 27%, and 7%, 16%, and 6% for the first and second scenarios at the Beibei Station, respectively.

- The suspended sediment in the studied region is generally sensitive to its lagged values rather than the lag discharge values. However, the MARS–KM models could estimate suspended sediment satisfactory using only discharge (Q) as inputs. It is very important in practical applications, because the measurement of suspended sediment is often very difficult.
- Comparison of models' ability in simulating cumulative suspended sediment loads also showed the superiority of MARS–KM compared to ANFIS, MARS, and M5Tree methods.

In this study, seasonality in the sediment load and discharge time series was not considered. It can be addressed in future studies. The methods may produce better estimates when season information is included in the models.

**Supplementary Materials:** The following are available online at https://www.mdpi.com/article/10.3390/su13094648/s1, Figure S1: Time variation graph of sediment load for (a) Guangyuan and (b) Beibei Station, Figure S2: Time variation graphs of the observed and estimated sediments by ANFIS, M5Tree, MARS and MARS-KM models in the test period at Guangyuan Station for the 1st training-test scenario, Figure S3: Time variation graphs of the observed and estimated sediments by ANFIS, M5Tree, MARS and MARS-KM models in the test period at Guangyuan Station for the 2nd training-test scenario, Figure S4: Time variation graphs of the observed and estimated sediments by ANFIS, M5Tree, MARS and MARS-KM models in the test period of Beibei Station—1st training-test scenario, Figure S5: Time variation graphs of the observed and estimated sediments by ANFIS, M5Tree, MARS and MARS-KM models in the test period at Beibei Station for the 2nd training-test scenario; Table S1: The statistical parameters of the data used in the study, Table S2: Performance of the best ANFIS, M5Tree, MARS and MARS-KM models in sediment prediction at Guangyuan Station, Table S3: Performance of the best ANFIS, M5Tree, MARS and MARS-KM models in sediment prediction at Beibei Station.

**Author Contributions:** Conceptualization: R.M.A., O.K., and K.S.P.; formal analysis: S.H., S.S., O.K., and R.M.A.; validation: R.M.A., S.H., K.S.P., S.S., and O.K.; supervision: S.S. and O.K.; writing original draft: R.M.A., S.H., K.S.P., S.S., O.K., and R.M.A.; visualization: R.M.A., S.H., and K.S.P.; investigation: R.M.A., S.H., and K.S.P. All authors have read and agreed to the published version of the manuscript.

**Funding:** This research was supported by the National Key R&D Program of China. (2016YFC0402706).

**Institutional Review Board Statement:** Not applicable.

**Informed Consent Statement:** Not applicable.

**Data Availability Statement:** The data presented in this study will be available on interested request from the corresponding author.

**Conflicts of Interest:** There is no conflict of interest in this study.

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
