# Peer review of "Suspended Sediment Modeling Using a Heuristic Regression Method Hybridized with Kmeans Clustering"

_sustainability, doi:10.3390/su13094648_

Round 1

Reviewer 1 Report

In my opinion, this article does not meet the requirements that an article published in the Sustainability journal should meet. Although the article is interesting and brings some new information, it is too extensive. The article is confusing and the information is not properly structured. In the first part there is only text and data, in the next part only graphs. I also consider the first picture to be very poor quality. For these reasons, I recommend returning the article for major revision.

Author Response

Thanks for your review of our study and giving us an opportunity to improve it. We tried to amend all queries and hope that changes satisfy the requirements.

Reviewer 2 Report

REVIEW:

Suspended sediment modeling using a heuristic regression method hybridized with Kmeans clustering

The manuscript deals with the comparison of several machine learning methods. The research herein presented is certainly within the scope of Sustainability.

According to my observations, the topic of the manuscript is interesting and challenging. However, the lack of clarity in some parts of the text should be addressed before the publication. I think the paper requires sharpening in the definition of the results obtained and subsequent discussion. Nonetheless, I am supportive with the manuscript and after the revision herein purposed I think it should be ready for publication. I will be happy to review an updated version of the manuscript.

List of comments

- Novelty needs to be highlighted in the abstract and introduction. References to previous works are provided but the new contributions need to be largely remarked.

- Line 121. There is typo just before the reference.

- Quality of Figure 1 is not admissible. Please, provide a non-blurry, high-quality figure.

- Section 2.1. Include a figure with daily data from Jan 1, 2007 to Dec 31, 2015. This helps the reader to visualize the time-series patterns.

- Section 2.1. A first comment deals with your dataset. At line 150 it is said that discharge and suspended information is collected on daily basis. Both sampling methods are essential for defining sediment fluxes at different time scales. It is therefore necessary to give more information on the monitoring and sampling methods (frequencies, location, method, managers...). It is then important to know (and prove) that the station managers have maintained a standardized protocol (for SSC and Q) throughout the period considered. Moreover, how often are these samples collected per day? How many samples are collected all through the study period? What is the distribution of the samples over the study period (are they evenly spaced in time to control grab samples, or were they mainly taken in a specific period, a research project...)?

- Section 2.1. Test and rupture tests are required before applying the machine-learning to evaluate the stationarity of your time-series (e.g. Mann-Kendall and Pettit tests). These analysis will allow the following results to be analyzed with more confidence and also it can detect any anomaly in the time series (dry or wet year within the study period).

- Section 2.2-2.5. Which software did you use for the implementation of these algorithms?.

- Section 3.1. Why did you analyze the sediment load instead of the suspended sediment concentration?. Is there any reason related with higher variability in the dataset or larger errors between model and dataset?.

- Section 3.2. Why didn’t the authors choose to split the time-series in seasonal periods?. The conclusions may be different.

- Section 3.2. Input combinations are not clearly outlined. Please, rephrase them, and explain carefully the acronyms Qt-1, Qt-3, etc.

- Tables 2-11 can be trimmed and incorporated as supplementary data.

- Sub-plots in Figure 10 need to be arranged.

- A discussion is missing. Authors should outline the strengths and weaknesses of data-driven methods. For instance, data-driven methods blindly fit the datasets. Consequently, sediment processes as sediment hysteresis are difficult to model adequately. This is an important point since sediment hysteresis helps to determine whether the river basin operates under supply-limited or available-limited sediment conditions. This is ultimately related to the aggradation/degradation processes occurring in the channel reach and it is important for river basin management. I would advise the authors to take a look at [1] and backup their results with the physical interpretation provided in [1].

- In the discussion section I am missing also a debate about the perspectives based on climate warming. Could the data-driven methods cope with time-series that are experiencing a temporal shift?.

Bibliography

[1] The origin of fine sediment determines the observations of suspended sediment fluxes under unsteady flow conditions. C Juez, MA Hassan, MJ Franca. WATER RESOURCES RESEARCH. 2018, 1-16.

Author Response

(The authors gave the same response as above.)

Reviewer 3 Report

sustainability-1172767-peer-review-v1

No

Lines

Comments

1

1-2

Title: The Title reflects the paper’s content accurately.

2

15-30

Abstract:

The abstract determines the paper’s content and objectives in a very manifest and complete fashion.

3

34-136

1.      Introduction

In L 39 a reference to Zarris et al [1] should be added and in L54 a reference to [2].

The increased stability aspect of MARS should be mentioned as in [3] and  that the k-means lack of robustness due to outliers is correctible as in [4][5][6].

Otherwise the introduction is both adequate and highly informative.

4

137-327

    2.  Materials and Methods

Quite well written and the case study selected is important.

5

328-568

3.      Application and Results

Well composed.

6

4.       Conclusions

Succinct and precise.

Concluding Remarks: It is a very interesting and well documented study which yields important results.

References

[1]      D. Zarris, M. Vlastara, and D. Panagoulia, “Sediment Delivery Assessment for a Transboundary Mediterranean Catchment: The Example of Nestos River Catchment,” Water Resour. Manag., vol. 25, no. 14, pp. 3785–3803, 2011.

[2]      D. Panagoulia, G. J. Tsekouras, and G. Kousiouris, “A multi-stage methodology for selecting input variables in ANN forecasting of river flows,” Glob. NEST J., vol. 19, no. 1, pp. 49–57, 2017.

[3]      J. H. Friedman, “Multivariate Adaptive Regression Splines,” Ann. Stat., vol. 19, no. 1, pp. 1–67, 1991.

[4]      A. Georgogiannis, “Robust k-means: A theoretical revisit,” Adv. Neural Inf. Process. Syst., no. Nips, pp. 2891–2899, 2016.

[5]      Y. Kondo, M. Salibian-Barrera, and R. Zamar, “RSKC: An R package for a robust and sparse k-means clustering algorithm,” J. Stat. Softw., vol. 72, no. 5, 2016.

[6]      Š. Brodinová, P. Filzmoser, T. Ortner, C. Breiteneder, and M. Rohm, “Robust and sparse k-means clustering for high-dimensional data,” Adv. Data Anal. Classif., vol. 13, no. 4, pp. 905–932, 2019.

Author Response

(The authors gave the same response as above.)

Reviewer 4 Report

The reviewer wants to thank the authors for their paper presenting a numerical study focusing on different advanced approaches for modelling of sediment transport. The reviewer has some (small) comments/questions/suggestions:

*1) Abstract: The presented values in the Lines (L) 26-28 are very specific but are not very useful for the reader. It is not clearly connected, which approach is connected to which value. Please either reduce this to a more general comment from x% to y% or clarify it.

*2) L37: SSL instead of SSl

*3) L55: This sentence would need a reference.

*4) L121: There is an issue with the reference.

*5) Section 2.1: Please clarify here which data / input values are later used and also present the sources of it.

*6) L150: The used time series ends in 2015. Why? The reviewer would assume that at least 5 more years should be possible, or not?

*7) Figure 1: the quality of the lower figure is not very good plus in both cases: is there a reference missing or are those figures made by the authors?

*8) Tab. 1: Please clarify the exact time span of each data set, training and test. Date from to Date.

*9) L287 and 288: Please correct the citation style. It should only be family name [xy].

*10) Section 3.1: The reviewer has to apologise but s/he got confused in this section and consequently it was hard to understand the following section. The (i) to (iv) in L331 to 333 are not identical with the ones mentioned in Table 2 and ongoing? Could you please clarify this.

*11) Table 2: Is the 1st train identical with the Training data from Table 1? If yes, why not use the same name. If not, why?

*12) There are two 3.2 sections.

*13) Figure 7 shows exactly what the reviewer would have asked: How good could maximum values be represented. Nevertheless, the cumulative sediments in Figure 10 provides a very good overview and relative comparison. But please use a grid a more meaningful axis. The graphs in Figure 10 are very important and need some attention to make it more readable.

The reviewer is looking forward to reading the paper again.

Author Response

(The authors gave the same response as above.)

Round 2

Reviewer 1 Report

This version is better.

Author Response

The language of the MS was carefully checked and corrected.

Reviewer 2 Report

Authors addressed all my queries listed in my previous review.

Author Response

(The authors gave the same response as above.)

Reviewer 4 Report

The reviewer thanks the authors for their corrections and answers. There are some small points, which should be still addressed.

*1) Section 3.1 and general: There are two training-tests 1st and 2nd. Normally one time period is used to get the model up and running and the quality is checked with an additional period. This should be clarified in this section.

*2) L 372 of the new version: please explain here or in a separate table exactly which combination is (i) to (vii).

*3) Figure 6: Thank you for the additional grid lines. Could you please make the x-axis more readable and use labels in the range of 12 or 24 month or switch to years. Plus there is a problem with the label in general: 1585 month = 132 years, really? Please check this again.

Thank you.

Author Response

All the comments were considered by the authors. Please see our responses in the attachment.
